# Utilising Digital Health Technology to Support Patient-Healthcare Provider Communication in Fragility Fracture Recovery: Systematic Review and Meta-Analysis

**DOI:** 10.3390/ijerph16204047

**Published:** 2019-10-22

**Authors:** Lalit Yadav, Ayantika Haldar, Unyime Jasper, Anita Taylor, Renuka Visvanathan, Mellick Chehade, Tiffany Gill

**Affiliations:** 1National Health and Medical Research Council Center for Research Excellence in Frailty and Healthy Ageing, Adelaide Medical School, The University of Adelaide, Adelaide, SA 5011, Australia; unyime.jasper@adelaide.edu.au (U.J.); renuka.visvanathan@adelaide.edu.au (R.V.); mellick.chehade@adelaide.edu.au (M.C.); 2Department of Orthopaedics and Trauma, Royal Adelaide Hospital, Central Adelaide Local Health Network, Adelaide, SA 5000, Australia; anita.taylor@sa.gov.au; 3Adelaide Medical School, Faculty of Health and Medical Sciences, The University of Adelaide, Adelaide, SA 5000, Australia; tiffany.gill@adelaide.edu.au; 4Aged and Extended Care Services, The Queen Elizabeth Hospital, Central Adelaide Local Health Network, Adelaide, SA 5011, Australia; ayantika.haldar@sa.gov.au

**Keywords:** digital health, telemedicine, health literacy, patient education, rehabilitation

## Abstract

The objective of this review is to evaluate the effectiveness of digital health supported targeted patient communication versus usual provision of health information, on the recovery of fragility fractures. The review considered studies including older people, aged 50 and above, with a fragility fracture. The primary outcome was prevention of secondary fractures by diagnosis and treatment of osteoporosis, and its adherence. This review considered both experimental and quasi-experimental study designs. A comprehensive search strategy was built to identify key terms including Medical subject headings (MeSH) and applied to the multiple electronic databases. An intention to treat analysis was applied to those studies included in the meta-analysis and odds ratio was calculated with random effects. Altogether, 15 studies were considered in the final stage for this systematic review. Out of these, 10 studies were Randomised controlled trials (RCT) and five were quasi experimental studies, published between the years 2003 and 2016 with a total of 5037 participants. Five Randomised control trails were included in the meta-analysis suggesting that digital health supported interventions were overall, twice as effective when compared with the usual standard care (OR 2.13, 95% CI 1.30–3.48), despite the population sample not being homogeneous. Findings from the remaining studies were narratively interpreted.

## 1. Introduction

In older people, a low energy trauma, such as a fall from a standing height or less, can result in a fracture. This is known as a “fragility fracture” and is usually due to osteoporosis with its associated reduced bone density [1]. One of the most devastating fragility fractures is the hip fracture. Due to the significant increase in ageing and life expectancy globally, it is estimated that the annual number of fragility hip fractures is likely to reach 6.3 million by 2050. Twenty percent of hip fractures can be fatal and a further 50% cause disability with only 30% of patients able to fully recover. [2,3,4]. Further, during recovery, older patients with a fragility fracture may exhibit sedentary behaviour and have low physical activity participation [5]. This adds to the complex scenario leading to a loss of independence, decreased mobility, and poor quality of life [6]. In parallel to this, the utilisation of health services and associated costs increases, mainly within the first year of the initial hospitalisation, much of which is attributable to ongoing long-term care [7,8,9,10]. Thus, this long-term care process requires an integrated approach that is often delivered by range of care providers, involves management of medication prescription for any coexisting comorbidities, exercise and falls prevention advice, good nutrition, and psychological well-being [11]. On the other hand, recent evidence in Australia suggests that the existing practice must also be applied to other types of fragility fractures that may not require acute hospitalisation or if it does, shorter periods of hospitalisation. Some of these fractures herald the beginning of the ‘fragility fracture cascade’ towards a hip fracture and include first occurrence of non-hip and non-vertebral minor fragility fractures [12]. 

Looking at the post-discharge care pathway; patients may need to attend orthopaedic out-patient departments (OPD), located in hospitals where access can be difficult, as they rely on family or ambulance services to provide transport. In addition, for falls prevention, patients may need to access specialist geriatric services and similarly, general practitioners (GP) within community for management of any existing co-morbidities and osteoporosis. For some, there may be involvement of community-based rehabilitation, aged care and allied health services. [13,14]. Shared decision making is enhanced through improved patient and family education where trust can be built, and people motivated to improve adherence to achievable treatment and prevention goals [11]. 

With the advancement of modern information and communications technologies (ICT), it is possible to integrate seamlessly the different service providers that are involved in the care of older people with fragility fracture and, more importantly, it could also include the patient and their nominated carers. Such technology may assist in the reorientation of services to the community and closer to the patient in their own community. Digital health or Electronic health (eHealth) systems utilise ICT built within computers, mobiles, sensors and web-based applications to support effective delivery of health services and information [15,16]. It involves people from multiple disciplines with expert knowledge and a desire to innovate; including specialties within health sciences, software engineering, communications, social science and legal aspects. However, the influence of human behaviour must be recognised, while interacting with the healthcare systems, as well as the local context in which the digital health platform is intended to be used [17]. This includes crucial components of health systems, such as financing, workforce, access to essential medicine and leadership and governance [15]. Importantly, three critical questions need to be addressed: (i) Does the information delivered through a digital health channel align with the recommended health practices or validated health content? (ii) Does the digital technology promise to achieve broader health sector objectives, as a discrete function? (iii) Are the software systems and communication channels able to facilitate the delivery of digital interventions and health content and demonstrate capability at scale? 

### Research Question

The question of this review is:

What is the effectiveness of digital health supported and targeted patient communication that is facilitated through healthcare providers in the recovery of older adults with fragility fractures? 

The review explores the use of different digital health strategies and its effect on treatment adherence, functional outcomes, quality of life, education, knowledge, and perceived service satisfaction.

## 2. Methods 

This systematic review was conducted in accordance with a standardised methodology for systematic reviews for effectiveness evidence [18] and this review title along with the methodological details has been registered with the Joanna Briggs Institute, The University of Adelaide systematic review database [19]. 

### 2.1. Inclusion Criteria 

#### 2.1.1. Participants

It is considered that in low- and middle-income countries, or other disadvantaged communities, the initiation of ageing processes could start at a relatively younger age. Therefore, the inclusion criteria involved original studies or research papers including people aged 50 and above with a low trauma or fragility fracture, and conducted within a hospital, residential aged care facility, or community dwelling. 

#### 2.1.2. Intervention

Studies were considered that evaluated digital health technology used to support targeted patient communication and education solutions delivered through any digital device in the form of voice call/message, text messages, educational videos or multimedia platforms (e.g., computers, mobile phone applications, telephone, and other audio-visual aids). 

#### 2.1.3. Comparators

Studies that compared usual provision of health information to patients delivered through instructions leaflet/booklet or any similar resources either at the point of discharge or as part of standard care.

#### 2.1.4. Outcomes

The primary outcome was prevention of secondary fractures by diagnosis and treatment of osteoporosis, and adherence to treatment. The secondary outcomes included quality of life, health/ehealth literacy, knowledge, or perceived service satisfaction. 

### 2.2. Types of Studies

This review considered both experimental and quasi-experimental study designs including Randomised controlled trials (RCTs), non-Randomised controlled trials, before and after studies and interrupted time-series studies. In addition, analytical observational studies including prospective and retrospective cohort studies, case-control studies and analytical cross-sectional studies were considered for inclusion. This review also considered descriptive observational study designs including case series, and individual cases. All studies published in English from the year 2000 until 2018 were included. 

### 2.3. Search Strategy

An initial limited search of MEDLINE was undertaken to identify articles on the topic. The text words contained in the titles and abstracts of relevant articles, and the index terms used to describe the articles were used to develop a full search strategy for PUBMED, CINAHL, SCOPUS, Embase, ProQuest dissertation and thesis global, and Google Scholar (Appendix A). The search strategy, including all identified keywords and index terms, was adapted for each included information source. The reference list of all studies selected for critical appraisal was screened for additional studies.

### 2.4. Study Selection

Following the search, all identified citations were collated and uploaded in EndNote X8/2018 (Clarivate Analytics, Philadelphia, PA, USA) and duplicates removed. Titles and abstracts were then screened by two independent reviewers (LY, AH) for assessment against the inclusion criteria for the review. Potentially relevant studies were retrieved in full and their citation details imported into the JBI System for the Unified Management, Assessment and Review of Information (JBI SUMARI) [20]. The full text of selected citations was assessed in detail against the inclusion criteria by two independent reviewers. Reasons for exclusion of full text studies that did not meet the inclusion criteria were recorded and reported in the systematic review. Any disagreements that arose between the reviewers at each stage of the study selection process were resolved through discussion, or with a third reviewer [TG]. The results of the search are reported in full in the final systematic review and presented in a Preferred Reporting Items for Systematic Reviews and Meta-analyses (PRISMA) [21]. A checklist is provided in Appendix A. 

### 2.5. Assessment of Methodological Quality

Eligible studies were critically appraised by the two independent reviewers (LY, AH) for methodological quality using standardised critical appraisal instruments for experimental and quasi-experimental studies from the Joanna Briggs Institute [18,20]. Any disagreements that arose were resolved through discussion, or with a third reviewer (TG). 

### 2.6. Data Extraction 

Data were extracted from included studies in the review using the standardised data extraction tool. The data included specific details about the populations, study methods, interventions, and outcomes of significance to the review objective. 

### 2.7. Data Synthesis 

Data from five studies were pooled in a statistical meta-analysis using JBI SUMARI. Effect sizes were expressed as odds ratios and heterogeneity was assessed statistically using I^2^ tests. Statistical analyses were performed using random effects models [22]. Whereas, statistical findings from the remaining ten studies included in the review were narratively interpreted. 

## 3. Results 

### 3.1. Study Inclusion 

Altogether, 3465 records were identified through database searching and additional 4 records from manual and secondary reference searches. After removing duplicates and articles with no clear orthopaedic or fragility/osteoporotic fracture or bone health domain, 1690 articles were screened for title and abstract. Further, 42 were considered for full text review, from which 15 studies were finally considered for this systematic review. The key reasons for excluding the studies were: digital health interventions targeting healthcare providers, other musculoskeletal conditions, falls prevention among older people without fragility fracture in the hospital setting, and digital health solutions aimed at health workforce education or meant for supporting clinical decision at the point of care. Out of the included studies, 10 studies were RCTs [23,24,25,26,27,28,29,30,31,32] and 5 were quasi experimental studies [33,34,35,36,37]; including 3 studies with no comparison group [33,35,37]. A PRISMA flowchart is provided as Figure 1. 

### 3.2. Methodological Quality

Overall methodological quality varied according to the type of studies. Within the RCT group, 2 studies were found to be of high-quality scoring 85% [28,29], 4 rated moderate, scoring between 50–70% [26,27,30,31] and 4 rated low at less than 50% [23,24,25,32]. The majority of studies scored poorly around questions relating to blinding or description of the intervention. 

Within the quasi experimental group, only one study scored around 90% [34] and two studies scored moderate [36,37]. Further, the remaining two scored low at less than 50% [33,35]. The reasons for low or moderate quality were linked to the absence of a control group and/or failure to clarify if follow-ups were completed. The results are summarised in Table 1.

### 3.3. Characteristics of Included Studies 

Studies included were published between the years 2003–2016 but none published in 2017 and 2018. The majority were conducted in Canada [24,25,27,28,29,31,35,37], followed by US [23,33,34,36] and one each in Australia, Italy and Thailand, respectively [26,30,32]. Participants were recruited from hospital-based settings in 12 studies [23,24,25,26,27,28,29,31,32,35,36,37] and community-based settings in 3 studies [30,33,34]. There were no studies conducted in the residential aged care facilities. A total of 5037 participants with fragility fractures were recruited to the 15 studies. Eight studies had only hip fractures as a criterion for inclusion [23,25,26,27,30,32,33,36] and 4 studies had women participants only [24,26,31,35]. 

### 3.4. Review Findings 

Findings of this review are categorised primarily into meta-analysis and narrative synthesis. Further, these results are presented in three subcategories (Table 2 and Table 3 (part A and part B)), which correspond to the definitions suggested by latest WHO guideline on digital health interventions [15]. 

### 3.5. Meta-Analysis

#### Targeted Patient Communication with Primary Care Physician Support

Five of the studies were included in the meta-analysis with a similar intervention strategy and measures of primary outcomes [24,25,28,29,31] (Table 2). Four of the studies involved voice telephone calls as a delivery channel and one [24] involved targeted patient communication using educational videos. Further, in all five studies, respective primary care physicians were also involved. Studies [24,31] having two separate intervention groups were combined as a single intervention as there was no reported difference between each intervention group, individually or combined and when compared with the control group. 

Findings from the studies included in this theme suggests that the primary outcome of bisphosphonate treatment for osteoporosis along with BMD test was significantly improved in the intervention group in comparison to the control group [25,28,29,31], except one study reported an increase with respect to osteoporosis treatment but not BMD testing [24]. However, when stratified by sex, men in the intervention group were less likely than women to receive appropriate secondary fracture prevention care at 15% and 44%, respectively [29]. Uptake of calcium and vitamin D improved in the intervention group [28,29] expect in one study [25]. Similarly, the intervention resulted in the majority of patients having a discussion about osteoporosis with their physician (82% vs. 55%, OR-3.8, 95% CI 2.3–6.3, *p* < 0.0001) [28]. 

This meta-analysis suggests the digital health supported interventions were twice as effective when compared with the usual provision of health information to patients as part of standard care (OR 2.13, 95% CI 1.30–3.48). This is statistically significant (z = 3.01, *p* = 0.003), despite the population sample not being homogeneous (I^2^ = 79, *p* = 0.005). The results are presented in Figure 2.

### 3.6. Narrative Synthesis 

#### Targeted Patient Communication

Studies included under this theme utilised digital health interventions in the form of educational videos and motivational voice telephone calls. These studies included 5 RCTs [23,26,27,30,32] and 2 quasi experimental [34,36] without involvement of primary care physicians as part of the intervention (Table 3, part A). Videos were used in two studies [23,36] whereas four utilized voice calls [26,30,32,34] and one study used both of these modes of delivery [27]. 

As an outcome, four studies reported exclusively around physical activity [27,30,32,36], and one each around functional status [23], adherence to prescribed treatment [34], proportion of falls [26], feasibility measured by recruitment and retention rate [27], health-related quality of life and anxiety and depression [30]. Physical activity measures varied across four studies. In one study, there was no difference between groups measured through the distance walked in feet; except some improvement in time (seconds) walked at 3-months post-discharge [36] whereas, another study reported improvement in the intervention group measured by daily steps and time spent walking [30]. Similarly, the third study also reported a significant increase in physical activity in the intervention group [32]. 

Functional status was assessed in one study using 36-item short form health survey (SF-36) at 6 months; no significant post-intervention differences were observed [23]. The study reporting adherence to osteoporosis treatment was effective in 70% of cases as compared to 46% in a representative population national survey [34]. No difference was found in the proportion of falls between two groups at 6 months [26]. However, another study demonstrated improvements in quality of life, anxiety and depression scores [30]. 

### 3.7. Telemedicine, Personal Health Tracking and Healthcare Provider Decision Support 

Multimedia applications were used in three quasi experimental studies with interactive telemedicine [33,35,37], but real time teleconsultation was provided only in one study [35] (Table 3, part B). Two studies reported good client or service satisfaction [33,37]. Interestingly, in one study, more than 50% of the patients never had any computer experience in their lifetime but successfully participated in the intervention [33]. However, from the outcomes measured (physical activity measured as hours per week, exercise self-efficacy, physical functioning, role-limitations due to physical health problem, social functioning, and health transition), there were no statistically significant differences observed [33]. The remaining studies reported positively around knowledge about osteoporosis and confidentiality issues [35] and other outcomes such as pain, shoulder range of motion (ROM) and upper limb function [37]. 

## 4. Discussion 

Fragility fractures usually affect older people and they require health care solutions which align with their daily needs and lives [38,39]. This review suggests digital health interventions can range from simple voice call to more sophisticated application of multimedia technologies to motivate and educate patients. Our review consisted of 15 studies including 10 RCTs and five quasi experimental with variation around methodological quality. Of these, there were three assessed as high, six of moderate and six of low methodological quality. The meta-analysis, included five studies, mostly conducted in the developed country setting except one study in Thailand [32]. Findings from the meta-analysis further suggest that digital health supported targeted patient communication with primary care physician involvement could be twice as effective as usual care in prevention of secondary fractures among patients with fragility fractures. In this review, secondary fracture prevention considered bone mineral density testing and osteoporosis treatment initiation and/or its adherence as surrogate endpoints [40,41]. Furthermore, narrative synthesis indicates there could be an improvement in secondary outcomes such as health-related quality of life, self-efficacy including physical mobility and physical activity. Thus, digital health can be incorporated in the design of a comprehensive solution, keeping patients and their carers at the centre. However, technology on its own is unable to work effectively unless key health systems challenges like information provision, availability and quality of services, acceptability to local practice and context, utilisation and efficiency of care provision are considered. Further, patient-side costs and community feedback mechanisms also need to be considered [15]. 

Our review findings suggest that a voice telephone call is effective during the follow-up recovery period, if provided by motivational and competent staff working with patients to resolve concerns or barriers. In one of the reviewed studies, where a more sophisticated computer system was used for physical exercises without intervention from a healthcare provider, ehealth literacy did not seem to be important in adhering to the intervention [33]. Technology has been used to devise a clinical decision support system, but whether this can be used to improve clinicians’ efficiency or for the purposes of task shifting [42,43] is unclear. There remains uncertainty around whether older patients are able to interact with this technology and optimise the benefits in their path to recovery from a disease condition [44]. The WHO report on healthy ageing suggests that ageing must be considered as a continuum of life and not stigmatised, suggesting that there is a decline in the intrinsic capacity of people as determined by the age bracket [38]. However, people within similar age brackets may have different intrinsic capabilities and health and social care services should be targeted to be efficient. With appropriate facilitation, patients can be empowered to utilise new technology and engage with the health system to form a credible information or knowledge exchange process with their health care providers or family and friends within their community [38,39]. For this to happen, we must engage all these stakeholders at different levels from policy to practice including patients and their carers to co-create a model of care using digital health technological solutions [45]. In addition, studies from our review also suggest future research must consider complex nature of clinical interventions in the area of orthopaedics or musculoskeletal issues with respect to ageing. These intervention approaches combine medical/surgical and/or psychosocial components. Thus, deploying range of patient engagement strategies involving their family, caregiver and social networks could help with treatment compliance [23]. Further, resources must be made available for long-term follow-ups [33,34] and larger investigations are required in a real-world setting to optimise delivery of patient education materials reflected through better retention and satisfaction rates [27].

Although, the majority of the studies included in our systematic review were RCTs, the overall methodological quality was not uniform. Some of the studies reported high attrition or small sample sizes [23]. One study was conducted as a RCT but was a pilot study to ascertain recruitment and retention rates, as a primary outcome variable [27]. The usual caveat is in the delivery process detail and must be interpreted carefully as each setting and context might differ [15]. As with conventional clinical trials, dose and response are determined but in cases like these several factors and confounders play a critical role [46]. Therefore, it is difficult to generalise findings and the results need to be interpreted with caution. The meta-analysis of relevant RCTs included an intention-to-treat analysis with random effects. This conservatively estimated the effect size of the intervention using one or more digital health solutions to communicate or educate patients with fragility fractures in order to support their recovery process. Quasi experimental studies were included, with the intention of understanding any novel approaches being tested. Such studies seem promising in order to achieve potential outcomes but as the sample sizes were based on convenience and relatively small, it is difficult to determine whether such solutions can be adopted into mainstream. Moreover, the majority of studies were conducted in developed countries so these findings cannot be generalised to low and middle income countries. 

In addition to these limitations, our meta-analysis was based on the assumption that the intervention components were broadly the same. Therefore, we might have neglected the dosing component with respect to how precise it would have been delivered to reach to the desired outcome compared. However, while interpreting, we acknowledge this fact and therefore recommend that any findings that suggest that interventions are effective (including our case of this meta-analysis) must be tested according to the local context. This will require adjustments for dosage and other aspects of technology suited for feasibility before going on a large-scale implementation. Finally, publication bias was not formally tested for as part of the meta-analysis. 

### 4.1. Recommendations for Practice 

Dedicated fracture liaison services and community based general practices managing patients with fragility fractures could be improved by implementing targeted digital health strategies adapted for local context.

### 4.2. Recommendations for Research 

Management of older people with fragility fractures is a complex, multidimensional problem which extends beyond the acute care facility and immediate discharge care. Patient recovery is often influenced by factors not related to bone fracture itself, this includes the individual’s intrinsic capability and external environment and social determinants. After the immediate discharge period, patients are often left on their own to navigate the health and care system leading to poorer post-fracture outcomes. Future research studies should be designed for fragility fractures with consideration of these factors. Digital health studies should also be undertaken using these principles applied to broader populations with other complex diagnoses requiring more person centred and integrated care. 

## 5. Conclusions

Findings from our review support the view that a person-centred and integrated model of care can be delivered to older people with fragility fractures with the support of digital health technological solutions and achieve desired outcomes. Resources to optimise pain management, physical activity, nutrition, sleep hygiene and mental health could all be integrated. The provision of health information in isolation does not equate to education. Monitoring and feedback of progress are critical. Techniques such as behaviour change and motivational interviewing need to be integral to the service. Importantly, solutions must be co-designed or co-created within the context of a particular practice and with consideration to resource availability to realise a model of care pathway that is fully feasible to implement in practice.

## Figures and Tables

**Figure 1 ijerph-16-04047-f001:**
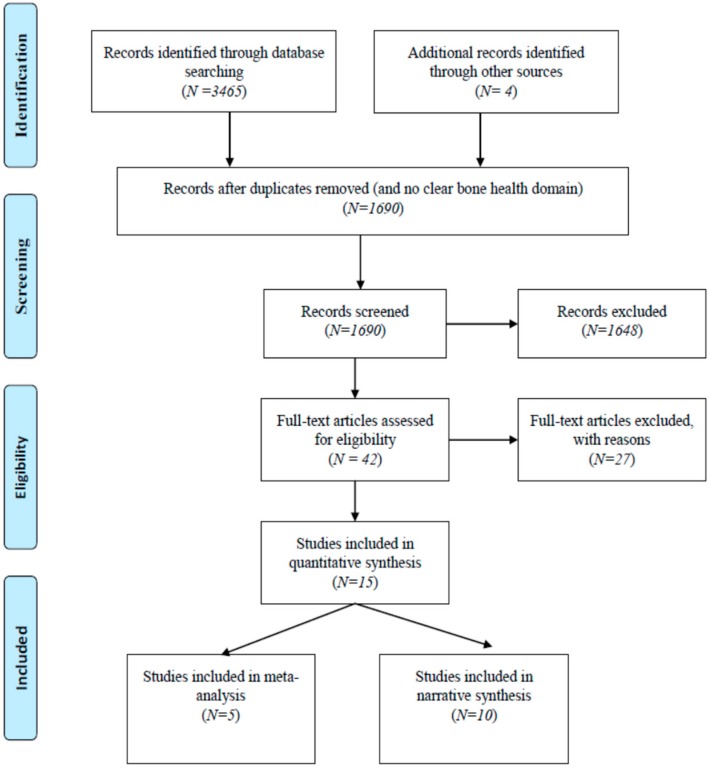
Preferred Reporting Items for Systematic Reviews and Meta-analyses (PRISMA) Flowchart.

**Figure 2 ijerph-16-04047-f002:**
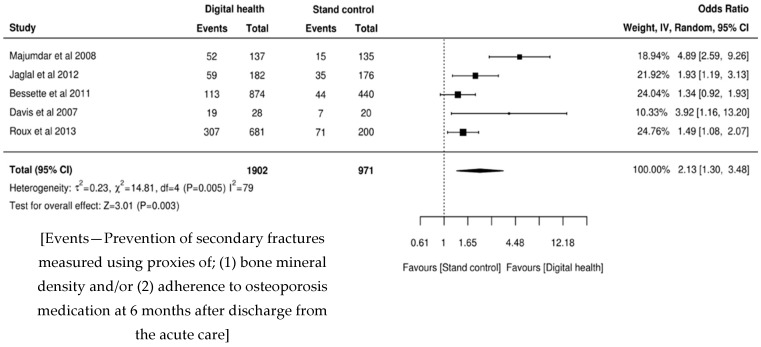
Forest plot.

**Table 1 ijerph-16-04047-t001:** Methodological quality assessment.

**1A Randomised controlled trials** (*N = 10*).															
**Study**	**Year**	**Q1**	**Q2**	**Q3**	**Q4**	**Q5**	**Q6**	**Q7**	**Q8**	**Q9**	**Q10**	**Q11**	**Q12**	**Q13**	**Overall Appraisal**
Allegrante et al. [23]	2007	Unclear	Yes	Unclear	Unclear	No	Unclear	Yes	No	No	Yes	Unclear	Yes	No	4 (31%)
Davis et al. [25]	2007	Unclear	Unclear	No	Unclear	No	Unclear	Yes	Unclear	Yes	Yes	No	Yes	Yes	5 (38%)
Majumdar et al. [29]	2008	Yes	Yes	Yes	No	Unclear	Yes	Yes	Yes	Yes	Yes	Yes	Yes	Yes	11 (85%)
Bessette et al. [24]	2011	Yes	No	Yes	No	No	Unclear	No	Yes	No	Yes	No	Unclear	No	4 (31%)
Jaglal et al. [28]	2012	Yes	Yes	Yes	Unclear	No	Yes	Yes	Yes	Yes	Yes	Yes	Yes	Yes	11 (85%)
Roux et al. [31]	2013	No	No	Yes	No	Yes	No	Yes	Yes	No	Yes	Unclear	Yes	Yes	7 (54%)
Suwanpasu et al. [32]	2014	Yes	No	Unclear	No	No	No	Yes	Unclear	No	Yes	Unclear	Yes	Yes	5 (38%)
Langford et al. [27]	2015	Yes	Yes	Yes	No	No	Yes	Yes	Yes	No	Yes	No	No	No	7 (54%)
Monaco et al. [26]	2015	Yes	Yes	Yes	No	No	Yes	Yes	Yes	No	Yes	No	Yes	Yes	9 (69%)
O’Halloran et al. [30]	2016	Yes	Yes	Yes	No	No	Yes	Yes	Yes	No	Yes	Unclear	Yes	Yes	9 (69%)
		70%	60%	70%	0%	10%	50%	90%	70%	30%	100%	20%	80%	70%	
**JBI critical appraisal checklist questions for RCTs**															
Q1	Was true randomisation used for assignment of participants to treatment groups?														
Q2	Was allocation to treatment groups concealed?														
Q3	Were treatment groups similar at the baseline?														
Q4	Were participants blind to treatment assignment?														
Q5	Were those delivering treatment blind to treatment assignment?														
Q6	Were outcomes assessors blind to treatment assignment?														
Q7	Were treatment groups treated identically other than the intervention of interest?														
Q8	Was follow up complete and if not, were differences between groups in terms of their follow up adequately described and analysed?														
Q9	Were participants analysed in the groups to which they were Randomised?														
Q10	Were outcomes measured in the same way for treatment groups?														
Q11	Were outcomes measured in a reliable way?														
Q12	Was appropriate statistical analysis used?														
Q13	Was the trial design appropriate, and any deviations from the standard RCT design														
	(individual randomisation, parallel groups) accounted for in the conduct and analysis of the trial?														
**1B Quasi-experimental studies** (*N* = 5).															
**SR No**		**Study**		**Year**	**Q1**	**Q2**	**Q3**	**Q4**	**Q5**	**Q6**	**Q7**	**Q8**	**Q9**	**Overall Appraisal**	
1	Tappen et al. [36]			2003	Yes	Unclear	Unclear	Yes	Yes	Unclear	Yes	Yes	Yes	6 (67%)	
2	Cook et al. [34]			2007	Yes	Yes	Yes	No	Yes	Yes	Yes	Yes	Yes	8 (89%)	
3	Dickson et al. [35]			2008	Yes	Unclear	Unclear	No	No	Unclear	Unclear	Unclear	Unclear	1 (11%)	
4	Tousignant et al. [37]			2014	Yes	Yes	Unclear	No	Yes	Yes	Yes	Yes	Yes	7 (78%)	
5	Bedra et al. [33]			2015	Yes	Yes	Unclear	No	Yes	Unclear	Yes	No	No	4 (44%)	
Response rate					100%	60%	20%	20%	80%	40%	80%	60%	60%		
**JBI critical appraisal checklist questions for Quasi-experimental studies**															
Q1	Is it clear in the study what is the ‘cause’ and what is the ‘effect’ (i.e., there is no confusion about which variable comes first)?														
Q2	Were the participants included in any comparisons similar?														
Q3	Were the participants included in any comparisons receiving similar treatment/care, other than the exposure or intervention of interest?														
Q4	Was there a control group?														
Q5	Were there multiple measurements of the outcome both pre and post the intervention/exposure?														
Q6	Was follow up complete and if not, were differences between groups in terms of their follow up adequately described and analysed?														
Q7	Were the outcomes of participants included in any comparisons measured in the same way?														
Q8	Were outcomes measured in a reliable way?														
Q9	Was appropriate statistical analysis used?														

**Table 2 ijerph-16-04047-t002:** Meta-analysis: (Targeted patient communication with primary care physician support *).

SR No	Author/Year Methodological Quality (H, M, L) **	Intervention	Outcome	Results
1	Majumdar et al. (2008) [29] H	The intervention consisted of three component intervention; firstly, brief telephonic counselling by an experienced registered nurse. These messages emphasise on at a high risk osteoporosis and future fracture, requiring bone mineral test followed by appropriate treatment through bisphosphonates or other alternative treatments like calcitonin, hormone replacement therapy, raloxifine. Beyond delivering these messages, the registered nurse also answered questions and allayed any concerned expressed by the patients about their treatment. The nurse also emphasised on the importance of speaking to their physician about their health condition. Secondly, patient specific reminder was sent to their respective physician with the same set of messages. Thirdly, summary of evidence-based actionable osteoporosis guideline, with endorsements from 5 local opinion leaders were sent to the physicians	Primary outcome was starting treatment with a bisphosphonate within 6 months after the fracture. This was measured using patient self-report and confirmed through dispensing records of local community pharmacies. There was 100% agreement between self-reporting and dispensing records. Secondary outcome was Bone mineral density (BMD) test and a composite measure of quality, referred to as guideline concordant or “appropriate care” defined as having undergone a BMD test and receiving bisphosphonate treatment if bone mass was low or osteoporosis.	Median age reported was 60 years (IQR 55–68 years). The findings from this study suggests that 22% (30) of the intervention group achieved primary outcome of bisphosphonate treatment for osteoporosis in comparison to only 7% (10) in the control group (RR 2.6, 95% CI 1.3–5.1, *p* = 0.008). By the end of the study, 66% received both calcium and vitamin D in the intervention group verses 43% with control group (RR 1.6, 95% CI 1.2–1.9, *p* = 0.001). Similarly, 52% (71) of the patients in the intervention group undertook BMD test compared to 18% (24) in control group (RR 2.8, 95% CI 1.9–4.2, *p* < 0.001). Of these, who had BMD test, 28% reported to have normal bone mass, 52% osteopenia and 20% osteoporosis at either hip or spine. Appropriate care was received by 38% of patients within the intervention group in compared to 11% in the control group (RR 3.1, 95% CI 1.8–5.3, *p* < 0.001). further, when the results were stratified by sex, men in the intervention group were less likely than women to receive appropriate care as 15% and 44% respectively.
2	Jaglal et al. (2012) [28] H	The intervention involved a physiotherapist as a centralised coordinator for following up patients and their physicians, provided evidence-based recommendations about the fracture risks and treatment of osteoporosis and assist with multidisciplinary consultation for patients with complex need, if needed. Patients received telephonic counselling about the risk of future fractures, BMD test and treatment of osteoporosis and follow-up with their physician. The primary care physician received a letter about the patient around the risk of future fracture, importance of BMD test, osteoporosis treatment using bisphosphonates or other alternative medications and availability of telehealth multidisciplinary consultation at a tertiary care hospital, in case of complex cases. Physicians also received pocket cards containing best-practice recommendations according to the recent Canadian guidelines. Outcomes	The primary outcome in was the proportion of patients self-reporting “appropriate management” defined as receiving within 6 months of fracture, either and osteoporosis medication (bisphosphonate, raloxifen or teriparatide) or normal BMD and prevention advise. Secondary outcomes were; the proportion of patients with a physician visit to discuss osteoporosis after fracture and the proportion for which BMD was scheduled or performed	Mean age was 66 years. The study reported a significant improvement in the osteoporosis management within the intervention group (32% vs. 20%, *p* = 0.007); analysis carried out as intention-to-treat. Further, in the intervention group, 23% had normal BMD, 22% treatment in comparison to 9% and 17% respectively in the control group. Whereas, with straight comparison, proportionately BMD test was reported to be higher in the intervention group than control group (57% vs. 21%, OR-4.8, 95% CI 3.0–7.0, *p* < 0.0001). Similarly, the intervention resulted in the majority of patients having a discussion about osteoporosis with their physician (82% vs. 55%, OR-3.8, 95% CI 2.3–6.3, *p* < 0.0001).
3	Bessette et al. (2011) [24] L	There were two intervention groups; written material and videocassette and written material group. The former received educational material on osteoporosis in the form of a two page document with concise information on the elevated risk for a new fracture, the importance of BMD and a summary of non-pharmacological therapies. Participants were invited to provide their respective primary care physician with an official summary of Canadian best practice guideline on osteoporosis. The videocassette group, in addition to written materials, received a 15-min educational video on osteoporosis which consisted of more comprehensive information on osteoporosis diagnosis and treatment as well as complications associated with fragility fractures	The primary objective of the study was to evaluate the impact of the two educational interventions on the diagnosis and treatment rates for osteoporosis after approximately 12 months following randomisation	In the group of women with no diagnosis or treatment at randomization, diagnosis of osteoporosis after follow-up occurred in 12%, 15%, and 16% of women within control, written material and videocassette and written material groups respectively. The rate of diagnosis for both intervention groups combined was 15%. Whereas, treatment rates were 8%,12% and 11% respectively in the same groups and if both interventions combined, it was 11%. In the group, without treatment at randomisation, at the follow-up, osteoporosis therapy was initiated in 10%, 13% and 15% respectively while combining the intervention; it would result in 13%.
4	Davis et al. (2007) [25] L	The intervention consisted of patient empowerment and physician alerting (PEPA) system; usual care for the fracture including surgical treatment, osteoporosis information and a letter for participants that encouraged them to return to their PCPs for further investigation, a request for participants to take a letter from the orthopaedic surgeon to the PCP alerting them to the hip fracture and encourage osteoporosis investigation, and telephonic call at 3 and 6 months to determine whether osteoporosis investigation and treatment had occurred	BMD test and bisphosphonate therapy at 6 months	In the PEPA (intervention) group, 15 (54%) were prescribed bisphosphonates therapy, 8(29%) BMD scan, 11(39%) calcium and vitamin D and 9(32%) exercises. Whereas, within the control group, none of the patients received any intervention, except 30% of them were prescribed calcium and vitamin D
5	Roux et al. (2013) [31] M	The intervention included two groups- Minimum (MIN) and Intensive (INT). MIN involved a coordinator to explain the patient, verbally and in writing, the casual link between fragility fracture and osteoporosis and the importance of contacting their primary care physician (PCP). A standard letter notified PCP of their patient’s fragility fracture status, explained the rationale and importance of rapid treatment of osteoporosis and outlined the appropriate investigations and treatment available and suggesting investigations and empirical treatments, irrespective of the BMD scores. Trained personnel made follow-up telephone calls at 6 and 12 months. In addition to collection of data, the importance of osteoporosis treatment was stressed and suggestions to increase adherence to osteoporosis medication was discussed. Whereas, within the INT group, same process was followed but in addition screening blood test were prescribed and patients were given a written prescription for BMD test. Blood test were conducted for serum calcium, phosphate, creatinine, alkaline phosphatase, and 25(OH) vitamin D levels, total blood count and plasma protein electrophoresis. Results were sent the respective PCPs with a letter stating that an incident fragility fracture usually indicates a need for treatment, irrespective of BMD results. When lab abnormalities were identified during screening, individualised counselling was given in writing to the PCP. Further, any PCP could contact one of the team members to discuss on how to manage the patient, if required. Similarly, telephonic follow-up was performed at 4, 8 and 12 months.	BMD test and osteoporosis therapy confirmed with the patients’ pharmacists at one year	Median age was 65 years (IQR 57–76 years) including 82% as females. At 12 months follow-up, the rates of current osteoporosis treatment were significantly higher in the intervention group than in the control group with no significant differences between the two intervention groups. Further, according to self-reports, around 45% of all patients underwent BMD testing during the first year, including 66% in the INT group whereas, around 34% within control and MIN groups each

* In this group, digital health technology was used to communicate with patients along with engagement of primary care physician through non-digital or conventional forms of communication. The primary care physicians were provided with patients’ key information around their disease conditions and future risks around bone health. Thus, such personalised information around their patients’ health status would encourage them to support decision making around appropriate investigation and treatment to prevent future falls. ** Methodological quality; H-High > 85%, M-Moderate 50–80%, L-Low < 50%.

**Table 3 ijerph-16-04047-t003:** Narrative synthesis.

**3A Targeted Patient Communication ***				
**SR No**	**Author/Year Methodological Quality (H, M, L) *****	**Intervention**	**Outcome**	**Results**
1	Allegrante et al. (2007) [23] L	Motivational videotape and a corresponding booklet around falls prevention self-efficacy, in addition an in-hospital peer support visit and 8-weeks out-patient physical therapy consisting of tailored exercises and progressive muscle strengthening training.	Functional status was assessed using 36-item short form health survey (SF-36) as the study’s primary outcome at 6-months follow-up	All the intervention patients were exposed to at least one of the three intervention components, i.e., videotape, strength training and peer counselling. However, only 34% of all the participants were able to complete full 6-months follow-up assessments (Intervention 32 vs. control 27). Patients within the intervention group had a significant positive change in the role-physical scale as compared to the control group (mean score, −11 ± 33 Vs −37 ± 41, *p* = 0.03). No significant post intervention differences were observed in the change on the physical functioning and social functioning scales and other domains like bodily pain, general health, vitality, role-emotional and mental health
2	Tappen et al. (2003) [36] M	Consisted of two parts; videotaping the study participants during their physical therapy sessions and showing one of the two generic educational videos that were produced for this study, depending upon the type of surgical repair, i.e., total hip replacement or arthroplasty, using plates or screws. Generic educational videos depict all aspects of physical recovery through the use of demonstrations and interviews with actual patients. The major focus of these generic tapes was the need to increase activity daily and intended to reinforce instructions that were given during rehabilitation and applied while moving in home or the respective community setting. Symptoms of anxiety and depression after hip surgery were also addressed using psychosocial adjustments. On the other hand, individual videos consisted of intervention participants being videotaped during their respective physical therapy sessions at regular intervals throughout their stay to record their progress. These videos show the therapist instructing the individual participants in the use of assistive devices, ambulance techniques and procedures for transferring. The tapes also document individualised instructions on exercises and show the therapist helping the participant do the prescribed exercises correctly. Participants were given both videotapes to take home for review	Physical activity performance measured through the distance walked in feet and time in seconds at three months post-discharge.	At three months post-discharge, time walked in seconds was significant, intervention [314.79 (SD-139.59)] Vs control [204.77 (179.70)].
				Though analysis comparing the two groups did not differ significantly on self-care, functional ability, coping and performance of independent ADLs; results for coping approached statistical significance.
3	Cook et al. (2007) [34] H	“Scriptassist” telephonic counselling program, intervention delivered telephonically by one of the four registered nurses at the scriptassist call centre. This communication was based upon the principles of motivational interviewing, focused on patient’s motivation for treatment, problem solving to resolve barriers to adherence, improve self-efficacy and helping patient practice skills to self-manage their own chronic conditions. Calls focused on relationship building and answering questions to encourage participant’s motivation for treatment.	Adherence to osteoporosis medication at 6 months based on pharmacy and clinical interview data	Among the high risk participants for fragility fractures, up to five telephonic contacts (median) were made with average call duration of 15 min. The participants were followed up for an average of 4.1 months after the start of the treatment (range 0–14 months). In terms of 6-months follow-up, 188 patients completed pharmacy data whereas, 255 patients with interview data. Adherence to treatment was reported at 6 months around 70% in this study, by both methods, compared to 46% in the representative population group reported through a national survey.
4	Monaco et al. (2015) [26] M	The intervention included at least 3 h during the stay of the patients, an occupational therapist to assess home hazards of falling based upon a standard checklist to determine future risk of falling and subsequent recommendations were provided. The patients also received a brochure describing falls prevention strategies. Further, geriatric evaluation was conducted for health optimisation and possibility of withdrawing medications in use which may increase the risk of falls and oral supplements of vitamin D and calcium were prescribed to continue after discharge from the hospital. Single telephonic call by an occupational therapist after discharge to check for environmental hazards, behaviour in ADL, use of assistive devices and reinforced targeted modifications to prevent falls.	Proportion of falls between two groups at 6 months	As an outcome measure, no differences were found in the proportion of fallers between the two groups (RR 1.06, CI 0.48–2.34)
5	O’Halloran et al. (2016) [30] M	Telephonic-based motivational interviewing eight times during the study participation period, lasting about 30 min per session and one call per week. The intervention was delivered by a trained physiotherapist in motivational interviewing. The intervention was designed to address issues associated with ambivalence about change in activity, such as beliefs about physical activity, low confidence and fear of falling which may prevent people after hip fracture from being more active.	Participants were asked to wear an accelerometer fitted to the thigh for a seven-day period at baseline and then again after the intervention phase, to measure the amount of physical activity they completed (ActivPal). Physical activity was recorded as the number of steps taken per day, the time spent walking per day and the time spent sitting or lying each day (sedentary behaviour). Secondary outcomes were health-related quality of life assessed through AQOL 8-D	Physical activity seemed to be improved in the intervention group compared with the control group, measured by daily steps and time spent walking; 26% and 22% respectively. Intervention group improved in mobility-related confidence but no difference observed with respect to mobility-related function. Further, the intervention group also demonstrated improvements in health-related quality of life (5.8, CI 1.2 to 10.4, *p* = 0.015), anxiety (−1.8, CI −3.0 to −0.6, *p* = 0.004) and depression scores (−3.7 CI −6.3 to −1.1, *p* = 0.010).
6	Suwanpasu et al. (2014) [32] L	Physical activity enhancing program (PEP) which composed of four phases, covered five sessions of implementation within seven weeks post-hip fracture surgery, but combined both phone calls and face-to-face interactions. Phase-1 assess existing self-efficacy, outcome expectations for physical activity and being ready to change physical activity. Phase-2 involved preparation for strengthening self-efficacy and outcome expectations offered through individual education and training in structural exercise and daily life physical activity and the benefits of regular behaviour, verbal encouragement by credible sources, seeing others experience and visual cueing (physical activity after hip fracture booklet, poster and flipbook), and short and long-term goal setting. Phase-3 included practice for strengthening self-efficacy and outcome expectations, involved everyday workouts of structural exercises and daily life physical activity, re-evaluating goal setting, self-monitoring and re-interpretation and control of unpleasant sensations associated with physical activity. Phase-4 involved evaluation of physical activity behaviour, including the energy expenditure of physical activity.	Information on physical activity was collected at 6-weeks after discharge.	At 6-weeks post-discharge, there was a significant increase in physical activity in the intervention group compared with the control group after controlling for pre-fracture physical activity with an effect size of 0.18 (<0.01). The amount of overall physical activity of the intervention group significantly increased by 961.37 MET/min/week over the control group. Physical activity was effective in 65% of the PEP (intervention) group. The ratio of efficiency (markedly effective and effective) induced by the PEP was higher than that induced by usual care (65% vs. 48%) and similarly ratio of markedly effective induced by the PEP was significantly higher than that induced by usual care 30% vs. 8%).
7	Langford et al. (2015) [27] M	The intervention included one hour in-hospital educational session with a trained health professional, using the hip fracture recovery manual and four educational videos. The content of this education program followed a standard format as guided by the manual but was individualised for each participant, including a description of the type of fracture sustained, how it was surgically fixated, red flags to watch out for during the recovery phase, an exercise program (home-based reducing both the rate and risk of future falls), practical information about future falls prevention, review of home safety and environmental hazards and mobility and recovery goal setting. The videos were viewed at the bedside using a tablet and headphone. Teach-back method was used which intends to clarify and check participants’ understanding of materials and ensured that the participants were able to provide a verbal summary of the education provided to them. After discharge, the trained health professional telephoned participants up to five times in the first 4 months following hip fracture to provide further encouragement, falls prevention information, coaching to remain active, problem solving skills, mobility goal setting, and advise to help participants maintain and increase their prescribed home exercises. The content of the sessions classified according to the CALO-RE (Coventry, Aberdeen and London-Redefined) taxonomy of behaviour change.	This study was a pilot study and the primary outcome of the trial was feasibility measured by recruitment rate and participant retention	The recruitment and retention rate of participants in the study was 42% and 90% respectively
**3B Telemedicine, personal health tracking and healthcare provider decision support ****				
**SR No**	**Author/Year**	**Intervention**	**Outcome**	**Results**
1	Bedra et al. (2015) [33] L	Home automated telemanagement (HAT) system to support individualised exercise program which consisted of a home unit, HAT server and a clinician unit. Home unit guides patients at home in routinely following their exercise program in a safe and effective way. The unit sends this information through a landline or wireless connection to the HAT information system. This system is able to monitor progress in terms of patient adherence and compare the results with the prescribed level of activities by the respective clinician. On the other hand, clinician unit can be any web-enabled devise. This system provides tailored feedback to the patients motivates them based on the behavioural profile and notifies clinicians. The system can further empower patients with self-paced interactive multimedia education on the major aspects of hip fracture rehabilitation program. This education module can be individualised to each patient’s specific needs and is based on the concepts of social cognitive theory.	Exercise self-efficacy, physical functioning, role limitations due to physical health problems, social functioning, health transition and client satisfaction at 30-day	Overall, 14 patients were recruited to test the telerehabilitation system at their homes. Mean age was around 77 (±9), More than 50% never had any computer experience in their lifetime. The telerehabilitation system was successfully used by the hip fracture patients at their home regardless of their socioeconomic or computer literacy background. At the end of 30-day telerehabilitation program, exercise self-efficacy (9 ± 1 vs. 6 ± 3, *p* = 0.01), physical functioning (71 ± 31 vs. 38 ± 27, *p* = 0.009), role-limitations due to physical health problems (17 ± 12 vs. 6 ± 10, *p* = 0.05), social functioning (85 ± 28 vs. 54 ± 31, *p* = 0.01), health transition (22 ± 18 vs 47 ± 40, *p* = 0.05) and client satisfaction (31 ± 0.46 vs. 27 ± 4, *p* = 0.04) apparently seems to have improved. Also, physical activity in terms of hours per week demonstrated a significant increase (31 ± 14 vs. 24 ± 14, *p* = 0.04). Adherence to the telerehabilitation over a 30-day program was reported to be 90% and above in most domains.
2	Dickson et al. (2007) [35] L	Involved setting up of network studio within the osteoporosis research department at a Women’s college hospital. The technology included a set-top videoconferencing system with a 27 inch television. Space was allocated for two desks; one for telehealth coordinator and one for health professionals to allow for them to move and demonstrate various exercises/activities to the patients during the consultation. Each healthcare professional individually consulted the patient via a telehealth, providing them with the same information they would receive in an in-person consult.	Knowledge about osteoporosis, confidentiality, and client satisfaction	The mean age was 56.5 years and the average length of telehealth consultation was around two hours and the length of follow-up was 15-min. The response rate to satisfaction survey questionnaire was 67%. Out of which, 58% rated telehealth consultation as excellent in comparison to in-person specialist consultation and 33% rated as a good experience. Almost all the participants expressed their intent to be using it again and recommend to their friends and family members. Prior to consultation, 73% described their knowledge about osteoporosis as fair and 27% as good but after consultation, only 10% described as fair, 30% good and 60% rated as excellent. When rating confidentiality, 83% patients felt completely comfortable discussing their health problems during their telehealth consultation
3	Tousignant et al. (2014) [37] M	The intervention based on a modular design, a generic platform was built, consisting of a videoconferencing unit to provide telerehabilitation program over eight consecutive weeks. The treatment program was delivered twice a day, every day, either supervised by a physiotherapist through telerehabilitation or unsupervised. Patients had two telerehab sessions in week 1, 3 and 5 and only one on weeks 2, 4, 6, 7 and 8.	Three outcome measures were evaluated; pain, shoulder ROM and upper limb function and additional satisfaction with health services received.	Each session lasted for about 30–45 min and divided into three parts; warm-up, treatment program and question period. The treatment program was adjusted for each patient according to the number of weeks, post fracture. Every exercise program involved four exercise types based on the orthopaedic physician’s specifications; stretching, pain control, active/active assisted ROM and muscle building. The physiotherapist also adjusted the progression of exercises according to the patient’s progress. Pain decreased significantly between pre and post intervention as indicated by the SF-MPQ score (difference of 10.6 ± 12.4, *p* = 0.003) and VAS (difference 26.3 ± 21.8, *p* = 0.001) which was greater than minimal clinically important difference (difference over 5/45 for total descriptors). The shoulder ROM difference was greater than the interrater minimal clinical difference (more than 5–10 degrees) for all. A difference of 42.1 ± 11.4 (*p* < 0) in the upper limb function was observed, greater than the minimal clinical difference (change over 15/100). Further, around 82% was the overall satisfaction, considered to be very good.

* A unidirectional or bidirectional communication initiated by health system customised according to an individual’s specific needs, results in “tailored or targeted client or patient communication” whereby message content is matched according to the needs and preferences of an individual. This consists of transmitting targeted health information to patients or client based on health status or demographics, which can include health education, behaviour change communication, client-centred messaging. In this theme, the delivery channels mainly consisted of voice telephone or calls by healthcare providers and educational videos. ** Telemedicine is defined as the provision of health-care services at a distance where patients and providers are separated. This could include consultation between remote client and healthcare providers, real-time telemedicine or interactive telemedicine, remote monitoring of client health or virtual monitoring or telemonitoring. Whereas, personal health tracking involves use of mobile application by clients, phone-based sensors, health records, wearables, web tools and other applications that allow to review and track their health status in terms of self-monitoring of health status and active data capture. Further, healthcare provider (HCP) decision support is defined as digitised job aids that combine an individual’s health information with the health-care provider’s knowledge and clinical protocols in order to assist HCPs in making diagnosis and treatment decisions. This involves supporting service delivery according to care plans, guidelines and protocols [33] *** Methodological quality; H-High > 85%, M-Moderate 50–80%, L-Low < 50%.

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
