# Peer review of "Utilising Digital Health Technology to Support Patient-Healthcare Provider Communication in Fragility Fracture Recovery: Systematic Review and Meta-Analysis"

_ijerph, 2019, doi:10.3390/ijerph16204047_

Round 1

Reviewer 1 Report

An inclusion criterion was that studies had to report on patients who had had a fragility or osteoporotic fracture. The primary outcome was “the prevention and of secondary fractures by diagnosis of and treatment of osteoporosis”. To be included patients had to have an osteoporotic fracture so how can the diagnosis of osteoporosis be a primary outcome? Has the primary outcome, the “prevention of secondary fractures” actually been addressed, other than by improved adherence to medication? The number of falls was not influenced by the intervention, and no data are presented on improvement in BMD or subsequent fracture. Figure 2: to what do the events refer? I was unable to extract the numbers presented in the figure from data presented under ‘Results’ in Table 2. The search strategies for the various databases are presented in Appendix 1. The list of search terms is comprehensive but omits ‘telehealth’, ‘e-health’, ‘m-health’ and ‘behaviour change’. As with all reviews there will be shortcomings in the search process. None of the searches included the term ‘mhealth’. In PubMed, it might be assumed that the Mesh term ‘telemedicine’ will include the entry term ‘mhealth’ this is not so. A search on ‘mhealth’ [tiab] NOT ‘telemedicine [mh] returns 2,361 papers. Searches were for papers in English “from the year 2000 until 2017”. A random selection of these were checked. The search for healthcare restructuring [tiab] is reported to return 34 papers. Twelve of these papers were published prior to 2000. The search for education [mh] AND hip fractures [mh] AND rehabilitation [mh] is reported as 41 papers. The search returns 44 papers, 9 of which are from before 2000. The Google scholar search, “fragility fracture” AND eHealth is reported as returning 89 papers. Repeating the search from 2000 to 2018 inclusive returns 74 resources, some of which are not in English. Restricting it to 2017 returns 59 resource, some of which are not in English. Changing ehealth to mhealth returns over 14,400 and changing it to telemedicine returns 262. The PubMed search was comprehensive with 55 separate searches. For the other five databases, only eight searches were conducted with limited search terms. The reason for this should be explained. No mention is made in the inclusion criteria as to whether included resources could be papers, book chapters, conference proceedings, abstracts, letters etc. Line 182: the reported number of papers should be 3465 and not 3645. Line 249: a standard land line might not meet the definition of digital health if it a standard analogue landline.

Author Response

Thank you very much for reviewing our manuscript. On behalf of all the authors, we greatly appreciate your complimentary comments and constructive suggestions. We have carried out the necessary edits/changes suggested and revised the manuscript accordingly.

Please find attached a point-by-point response to reviewer’s comments. We hope that you find our responses satisfactory or any suggestion for further improvement of the manuscript.

Reviewer 2 Report

Overall good report. Conclusions are supported by similar literature for use of digital health care tools for chronic disease management.

Author Response

We really appreciate for your time and efforts in reviewing this manuscript. Thank you so much

Reviewer 3 Report

The paper deals with a digital health technology aimed at improvement of healthcare service for patient. It is rather a review of the other studies than the research paper. Although, it comprises a meta-analysis as the main analytical approach.

The abstract is very long. According to the instructions for the authors of the International Journal of Environmental Research and Public Health, abstract should contain a maximum of 200 words.

The Introduction section offers a general overview of the examined topic. The end of the chapter is covered by the research questions. Why is plural applied there? There is only one research question.

The Methods section contains a suitable description of the steps carried out in order to perform the desired analysis.

The subheadings used in the Meta-analysis section should be reformatted. Now they look like common text without a full stop at the end of the sentence. It is misleading.

The analysis outcome is not appropriately explained – for instance, the test statistic values on the line 238, 241, and 242 are nowhere described in the methodology. Through what type of analysis is odds ratio computed?

It is more common that the Discussion section contains the recommendations for practice or further research rather than the Conclusion section. Also, it could be more beneficial for reader to have it compared with the other studies within just right the discussion.

There are some grammar errors – for instance:

line 186: semicolon instead of colon; line 215: a space missing between “the” and “15”; lines 173 and 177: “Data was…” and “Data from five studies were…” – it has to be unified; line 271: “a third study” instead of “the third study”.

The text should be proofread. Also, it contains a mixture of the British English and the American English and therefore, it has to be unified into the particular type.

Author Response

Thank you very much for reviewing our manuscript. On behalf of all the authors, we greatly appreciate for your complimentary comments and constructive suggestions. We have carried out the necessary edits/changes suggested and revised the manuscript accordingly.

Please find attached a point-by-point response to reviewer’s comments. We hope that you find our responses satisfactory or any suggestion for further improvement of the manuscript.

Reviewer 4 Report

Dear Authors,

Thank you for your work in this very important and challenging topic.  You have done a nice job of reviewing studies, summarizing the results, and presenting your findings.  This is not a simple task and exposes even more research questions that it answers.  

Author Response

We really appreciate for your time and efforts in reviewing this manuscript. Thank you so much.

Round 2

Reviewer 1 Report

Thank you for opportunity to re-review this paper. Several queries and concerns remain.

Point 1:

Previous comment

An inclusion criterion was that studies had to report on patients who had had a fragility or osteoporotic fracture. The primary outcome was, “…the prevention and of secondary fractures by diagnosis of and treatment of osteoporosis”. To be included patients had to have an osteoporotic fracture so how can the diagnosis of osteoporosis be a primary outcome?

Response:

Thank you for your constructive comments and suggestions.

We appreciate this as a valid point and therefore the inclusion criteria specifies that the studies included in the review consisted of patients having low trauma injury or falls sustaining fragility fractures. Please refer to the “Participants” heading under methods section on page 3 of the manuscript for the corrections in bold in the statement as follows;

“Therefore, the inclusion criteria involved original studies or research papers including people aged 50 and above with a low trauma or fragility fracture, conducted within a hospital, residential aged care facility or community dwelling”.

New query:

The problem has not been resolved. The primary outcome as stated in lines 109-110 remains, “…the prevention and of secondary fractures by diagnosis of and treatment of osteoporosis. No diagnosis is reported and neither is a reduction in secondary fractures. As this was a registered systematic review, I am not sure as to how changing the inclusion criteria to, “Therefore, the inclusion criteria involved original studies or research papers including people aged 50 and above with a low trauma or fragility fracture, conducted within a hospital, residential aged care facility or community dwelling” resolves the problem of not addressing the stated outcome measures.  Has the primary outcome as stated in the registered review been met in the study? See below.

Point 2: Has the primary outcome, the “prevention of secondary fractures” actually been addressed, other than by improved adherence to medication? The number of falls was not influenced by the intervention, and no data are presented on improvement in BMD or subsequent fracture.

Response: Diagnosis and treatment of osteoporosis and its adherence has been considered as a surrogate to prevention of secondary fractures. We acknowledge that true number of falls was not considered as a primary outcome in the included studies, except in one study [26] which was considered in narrative synthesis in Table 3 (page 12).

New query: This has not been clearly stated in the paper. See comment below.

Point 3: Figure 2: to what do the events refer? I was unable to extract the numbers presented in the figure from data presented under ‘Results’ in Table 2.

Response:

Thanks for this comment. This has now been clarified in figure 2 on page 9, “Events- prevention of secondary fractures measured using proxies of; 1) bone mineral density and/or 2) adherence to osteoporosis medication at 6 months after discharge from the acute care”

New query: Events are not prevention of secondary fractures: a secondary fracture is an event. The Concise Oxford Dictionary defines and event as: a thing that happens or takes place. If it has not happened it is not an event. No reduction in secondary fractures was documented. What is being reported appears to be adherence to treatment and possibly the effect of this on improvement on bone mass density.

Point 4:

The search strategies for the various databases are presented in Appendix 1. The list of search terms is comprehensive but omits ‘telehealth’, ‘e-health’, ‘m-health’ and ‘behaviour change’. As with all reviews there will be shortcomings in the search process. None of the searches included the term ‘mhealth’. In PubMed, it might be assumed that the Mesh term ‘telemedicine’ will include the entry term ‘mhealth’ this is not so. A search on ‘mhealth’ [tiab] NOT ‘telemedicine [mh] returns 2,361 papers. Searches were for papers in English “from the year 2000 until 2017”. A random selection of these were checked. The search for healthcare restructuring [tiab] is reported to return 34 papers. Twelve of these papers were published prior to 2000. The search for education [mh] AND hip fractures [mh] AND rehabilitation [mh] is reported as 41 papers. The search returns 44 papers, 9 of which are from before 2000. The Google scholar search, “fragility fracture” AND eHealth is reported as returning 89 papers. Repeating the search from 2000 to 2018 inclusive returns 74 resources, some of which are not in English. Restricting it to 2017 returns 59 resource, some of which are not in English. Changing ehealth to mhealth returns over 14,400 and changing it to telemedicine returns 262.

Response:

We really appreciate for your efforts in looking at these additional searches. We followed the suggested searches to locate for any additional records. In addition confirming that MeSH term “Telemedicine” includes terms like telehealth, mhealth, ehealth as entry terms. However, we couldn’t identify relevant articles that could fit the inclusion criteria.  

We have attempted a comprehensive search reflecting terms like telemedicine, telehealth, mhealth, ehealth and behaviour change with respect to fragility fractures sustained due to low trauma among older people. Nevertheless, we acknowledge there might be some shortcomings in our search process offering as one of the study limitations.

New query: the response merely states that the authors could not find any more relevant articles. For any review, the search needs to be replicable. If more articles were identified, whether they were relevant or not, they should be rflected in the PRISMA figure.

Point 5: Line 249: a standard land line might not meet the definition of digital health if it a standard analogue landline.

Response:

Further, the statement in the line 249 (initial version of the manuscript) has been corrected as “Studies included under this theme utilised digital health interventions in the form of educational videos and motivational telephone or voice calls” on page 9 under heading “Targeted patient communication”. We acknowledge there was no mentioning of standard landline or standard analogue landline in any of the articles.

New query: line 201 refers to, “…telephone (landline) of mobile  based delivery channel…”.

Line 225 refers to “..telephone or voice calls...”,  what is the difference?

Line 284 refers to “… a simple telephone call…”

The title of the paper is, “Using Digital Heath Technology…” If cited papers in a systematic review are not using digital communication i.e. POTS -  ‘plain on telephones’, then they are not relevant. If this cannot be determined from their content, they should not be in the review. This needs to be resolved.

Point 6

New Query: There appear to be problems with the citation numbers in Tables 2 and 3. The reference numbers given in line 226 do not match those given in table 2. Table 3 is not referred to in the text.

Reviewer 3 Report

The corrections are made as it was proposed.

Author Response

COMMENT (ROUND 2):

The corrections are made as it was proposed

Response to Round 2 comment;

Thank you very much for reviewing our manuscript and providing the feedback. We once again really appreciate for your valuable time and effort.